# Inverted valley polarization in optically excited transition metal dichalcogenides

Gunnar Berghäuser[1], Ivan Bernal-Villamil[1], Robert Schmidt [2], Robert Schneider[2], Iris Niehues[2], Paul Erhart [1], Steffen Michaelis de Vasconcellos [2], Rudolf Bratschitsch [2], Andreas Knorr[3] & Ermin Malic[1]

Large spin–orbit coupling in combination with circular dichroism allows access to spin-polarized and valley-polarized states in a controlled way in transition metal dichalcogenides. The promising application in spin-valleytronics devices requires a thorough understanding of intervalley coupling mechanisms, which determine the lifetime of spin and valley polarizations. Here we present a joint theory–experiment study shedding light on the Dexter-like intervalley coupling. We reveal that this mechanism couples A and B excitonic states in different valleys, giving rise to an efficient intervalley transfer of coherent exciton populations. We demonstrate that the valley polarization vanishes and is even inverted for A excitons, when the B exciton is resonantly excited and vice versa. Our theoretical findings are supported by energy-resolved and valley-resolved pump-probe experiments and also provide an explanation for the recently measured up-conversion in photoluminescence. The gained insights might help to develop strategies to overcome the intrinsic limit for spin and valley polarizations.

[1] Department of Physics, Chalmers University of Technology, Gothenburg, 41296 Sweden. [2] Institute of Physics and Center for Nanotechnology, University of Münster, Münster, 48149 Germany. [3] Institut für Theoretische Physik, Technische Universität Berlin, Berlin, 10623 Germany. Correspondence and requests for materials should be addressed to G.B. (email: gunbergh@chalmers.se)

The promising implementation of monolayer transition metal dichalcogenides (TMDs) in spin-valleytronics is based on the spin-selective and valley-selective excitation of excitonic states. An important prerequisite for these applications is a profound understanding of the dynamics of optically excited spin and valley polarization. The challenge is to be able to control many-particle mechanisms by coupling and mixing excitonic states of different spin and valley. In previous theoretical and experimental studies, it has already been shown that in addition to relatively slow intervalley spin-flip scattering processes[1–5] there is an efficient intervalley coupling mechanism via Coulomb exchange processes[6–15]. This Coulomb-induced dipole–dipole interaction couples resonant excitonic states in K and K′ valleys, giving rise to a decay of valley polarization on a picosecond time scale. However, the coupling is relatively small and only occurs as a second-order process, since it requires a non-zero center-of-mass momentum that first has to be generated by exciton-disorder coupling[14], biexcitonic excitations,[15] or exciton-phonon coupling[16].

In this work, we reveal a new intrinsic and direct Coulomb-driven intervalley coupling mechanism that is much more pronounced than the Coulomb exchange term and determines the optically accessible spin-valley polarizations in TMDs. We show that this interaction resembles the Dexter coupling between two spatially separated systems[17,18]; however, now coupling different valleys in momentum space. It gives rise to an efficient intervalley transfer of coherent exciton populations between excitonic states with the same spin in different valleys. The theoretical predictions are supported by measurements of the valley polarization in energy-resolved and valley-resolved femtosecond two-color pump-probe experiments, and also provide a microscopic explanation for recently published photoluminescence experiments measuring an up-conversion from A to B excitons in the opposite valley[19].

## Results

**Theoretical model**. The theoretical approach is based on the density matrix formalism, providing microscopic access to the time-resolved and energy-resolved dynamics of microscopic quantities, such as the microscopic polarization $p^{ij} = \langle a_j^\dagger a_i \rangle$, which is a measure for the transition probability between the states $i$ and $j$[20]. We use the picture of second quantization introducing electron creation and annihilation operators $a_j^\dagger$ and $a_i$. The states are described by the compound indices $i, j$ including the electronic momentum $\mathbf{k}$, the spin-valley index $\xi_s$ and $\overline{\xi}_s$ denoting opposite valleys with the same spin, and the electronic band index $\lambda = v, c$ standing for the valence and the conduction band. First, we define the many-particle Hamilton operator consisting of the interaction-free carrier contribution $H_0$, the light–matter interaction $H_{l-m}$, and the Coulomb interaction $H_C$. Then, we derive the TMD Bloch equation for the microscopic polarization $p^{ij}$ by exploiting the Heisenberg equation of motion[21]. This quantity couples to incoherent carrier densities $\rho^{ii} = \sum_j \left( |p^{ij}|^2 + \Delta N_x^{ij} \right)$ including coherent $\left( |p^{ij}|^2 \right)$ and incoherent $(\Delta N_x^{ij})$ excitons[16,20,21]. The microscopic polarization is directly excited via an external light source that introduces the coherent excitons, which then decay via phonons into incoherent excitons[16]. In this work, we focus on the coherent exciton dynamics, where we include dephasing and the formation of incoherent excitons via phonon-scattering constants calculated on a microscopic footing[22]. A microscopic description of the dissipation dynamics of incoherent excitons into dark states away from the light cone is beyond the scope of this analysis[22]. Here we

have included the incoherent exciton dynamics within a Lindblad formalism[23] cf. Methods section.

The full TMD Bloch equations for the microscopic polarization and the incoherent excitons can be found in Eq. (3) in the Methods section. The equation includes a term describing the intrinsic Coulomb-induced intervalley coupling of excitonic states in different valleys (last line in Eq. (3)). This interaction is formally analog to the well-known Dexter interaction[17]; however, now coupling states are in the same material but in different valleys. In contrast to the exchange term, no center-of-mass momentum transfer between the valleys is necessary. Furthermore, since carriers in valence and conduction band carry the same geometric phase, the Coulomb-induced intraband exchange of carriers between different valleys conserves the total angular momentum. Therefore, the process does not violate the three-fold rotational symmetry of the system, leading to valley-selective circular dichroism.

The corresponding Dexter-like matrix element reads

$$D^{nm\xi_s\overline{\xi}_s} = \sum_{\mathbf{k},\mathbf{k}'} \theta_{\mathbf{k}}^{n\xi_{s*}} \theta_{\mathbf{k}'}^{m\overline{\xi}_s} \frac{C_{\mathbf{k},\mathbf{k}'}^{cc,\xi\overline{\xi}} C_{\mathbf{k}',\mathbf{k}}^{vv\overline{\xi}\xi} V_{\mathbf{k}-\mathbf{k}'+\Delta_{\xi\overline{\xi}}}^{2D}}{\varepsilon(\mathbf{k} - \mathbf{k}' + \Delta_{\xi\overline{\xi}})}. \tag{1}$$

It is inversely proportional to the valley distance $\Delta_{\xi\overline{\xi}} = K - K' = 4\pi/3a_0$. Although the valley separation in the momentum space is large, the Dexter coupling is still very important due to the remarkably strong Coulomb interaction in TMDs[24–33]. Note that the distance between the valleys $\Delta_{\xi\overline{\xi}}$ enters the Coulomb potential, while it does not effect the excitonic wave functions $\theta_{\mathbf{k}}^{n\xi_s}$ and $\theta_{\mathbf{k}'}^{m\overline{\xi}_s}$. Furthermore, in the vicinity of the K valleys the coefficients $C_{\mathbf{k},\mathbf{k}'}^{cc,\xi\overline{\xi}} C_{\mathbf{k}',\mathbf{k}}^{vv\overline{\xi}\xi}$ are given by a phase, which cancels the geometric phase of the excitonic wave functions[34], cf. Methods section for details on the applied theory. We can estimate the coupling analytically by exploiting $\left|\Delta_{\xi\overline{\xi}}\right| \gg |\mathbf{k} - \mathbf{k}'|$, which allows us to evaluate the sums over the excitonic wave functions using two-dimensional 1 s excitonic functions. Assuming an excitonic Bohr radius of 1 nm we find $\sum_{\mathbf{k}} \theta_{\mathbf{k}}^{1s\xi_s} = \theta^{1s\xi_s}(r = 0) = \sqrt{\frac{2}{\pi}} \frac{1}{a_B} \approx 0.6\,\mathrm{nm}^{-1}$. Finally, assuming the TMD monolayer on a borosilicate substrate that is characterized via a high-frequency dielectric constant of $\varepsilon = 4.6$ and exploiting that for large $q$ the internal many-particle screening is negligibly small[35], we can estimate the coupling strength to be in the range of 50 meV reflecting the high importance of the Dexter-like mechanism.

Before evaluating the full TMD Bloch equations (Eq. (3) in the Methods section), we first consider a simplyfied model to illustrate the efficiency of the initially off-resonant coupling. The Dexter interaction couples states with the same spin and in different valleys, i.e., A (B) excitons in the K valley couple to B′ (A′) excitons in the K′ valley, cf. Fig. 1. The nature of the Dexter coupling can be understood from a simple example: we assume two states with the eigenenergies $E_1$ and $E_2$ in different valleys. Valley 1 is optically excited with the Rabi frequency $\Omega_1(t)$ giving rise to the microscopic polarization $p_1(t)$, which then interacts with the other valley via a coupling constant $C_{12}$, inducing a polarization $p_2(t)$ in the unpumped valley 2. In Fourier space, the corresponding simplified Bloch equation can be solved analytically resulting in $p_1(\omega) = \frac{\Omega_1(\omega)}{E_1 - \hbar\omega - |C_{12}|^2/(E_2 - \hbar\omega)}$ for the optically excited valley and $p_2(\omega) = \frac{C_{12} p_1(\omega)}{E_2 - \hbar\omega}$ for the unpumped but indirectly driven valley. We can now express the relative energy-dependent polarization difference as $r_v^1(\omega) = \frac{p_1(\omega) - p_2(\omega)}{p_1(\omega) + p_2(\omega)} = \frac{1 - v_{21}(\omega)}{1 + v_{21}(\omega)}$, where we have introduced the ratio

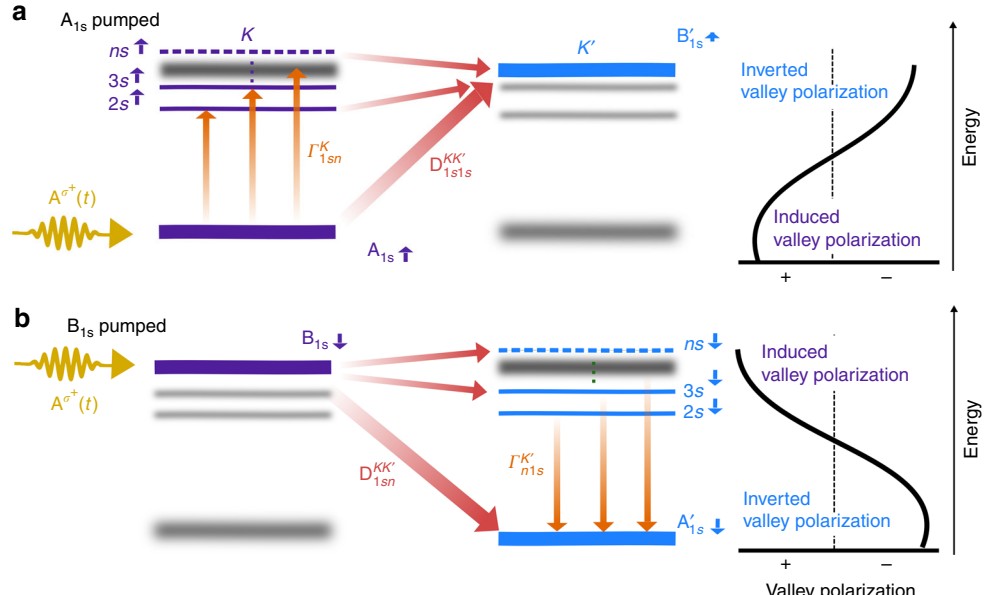

**Fig. 1** Schematic illustration of Dexter-like intervalley coupling. The coupling of **a** spin-up and **b** spin-down states after resonant excitation of $A_{1s}$ and $B_{1s}$ exciton in the K valley, respectively. The applied circularly polarized light is denoted by the vector potential $A^{\sigma+}(t)$. The states of the corresponding opposite spin are blurred in the background. Higher excitonic B states are not shown for reasons of clarity, but they have been considered in the calculations. The optically excited 1s state couples to higher excitonic states with the same spin in the same valley via the inter-exciton coupling $\Gamma_{1sn}$. At the same time an intervalley Dexter-like coupling $D_{1sn}^{KK'}$ induces an intervalley oscillation transfer between A and B excitons generating a microscopic polarization in the unpumped K' valley. This induces a frequency-dependent inversion of the optically induced valley polarization. Note that we denote the indirectly driven A and B excitons in the K' valley as A' and B' excitons

between the indirectly and directly driven valley as $v_{21}(\omega) = \frac{p_2(\omega)}{p_1(\omega)} = \frac{C_{12}}{E_2 - \hbar\omega}$. For resonant excitation to $E_1$ ($\hbar\omega = 0$), we find that the reduction of the valley polarization is determined by $\tilde{v}_{21} = v_{21}(\omega = 0) = \frac{C_{12}}{E_2 - E_1}$, i.e., the ratio of the coupling constant and the energy separation of the states, e.g., analog to the spin–orbit coupling in TMDs. In contrast, for resonant excitation to $E_2$, we find that the valley polarization is completely inverted $r_v^1(\omega) \to -1$. This is a surprising result as the state $E_2$ is only indirectly driven. The intervalley coupling leads to an energy-dependent dominance of the indirectly driven polarization. For the investigated exemplary material tungsten disulfide ($WS_2$), $A_{1s}$ and $B_{1s}$ excitonic states are separated due to the spin–orbit coupling by $E_1 - E_2 \approx 400$ meV. Using the approximated value of 50 meV for the Dexter coupling, we obtain a scaling factor of $\tilde{v}_{21} \approx 0.125$. This estimation suggests that when resonantly pumping the $A_{1s}$ state, approximately $1/(1/\tilde{v}_{21} + 1) \approx$ 11% of the light field is indirectly absorbed by the $B_{1s}$ state in the unpumped valley due to the Dexter coupling. This means that the induced valley polarization at the A exciton is decreased by 11% and inverted at the energy of the B exciton, where the Dexter coupling leads to an intervalley up-conversion. In the case of $MoS_2$, where the energy separation of $A_{1s}$ and $B_{1s}$ is in the range of 150 meV, the same estimation predicts an indirect absorption of 30% at the energy of $A_{1s}$ and $B_{1s}$ excitons in the unpumped valley suggesting a crucial role of Dexter coupling, c.f. Supplementary Note 4 for details on the estimation for different TMDs and substrates, portrayed in Supplementary Fig. 6. Note also that the discussed simple scaling model only takes into account the 1s states and therefore presents a lower limit for the Dexter coupling. Taking into account all possible coupling excitonic states as well as non-linear effects, Dexter coupling turns out to be even more efficient, as discussed below.

Now, we model ultrafast pump-probe experiments by evaluating the Bloch equation (Eq. (3) in the methods section)

for a non-linear excitation. We calculate the temporal evolution of the coherent and incoherent exciton densities in both valleys and for both spin systems, and evaluate the corresponding exciton population $N_s^\xi$ consisting of coherent and incoherent excitons for different delay times. Here $N_n^\xi$ describes the total exciton population in the valley $\xi$ and the excitonic state $n$. This allows us to compare the spin and valley polarization as a function of excitation energy, pump fluence, and delay time.

**Experimental methods.** To address the same quantities in experiments, we measure transient differential transmission of monolayer $WS_2$ with energy-resolved and valley-resolved femtosecond pump-probe experiments. A $WS_2$ monolayer on a borosilicate substrate is pumped by left circularly polarized laser pulses (200 fs), creating a population of excitons in a specific valley (K or K'). To probe the temporal and spectral dynamics of excitons we use linearly polarized pulses with a spectral bandwidth of 250 meV covering either the spectral range of A or B excitons. By analyzing the linearly polarized probe pulses transmitted through the monolayer for their circularly polarized components, we can simultaneously measure the dynamics in the pumped ("same circular polarization", SCP) and unpumped valley ("opposite circular polarization", OCP), cf. Supplementary Fig. 1. Observed changes of the spectral shapes at positive delay times are due to shifts of the exciton due to Coulomb renormalization, in particular biexcitons[15], and bleaching of the exciton absorption due to Pauli blocking[14]. Integrating the spectra in the range of the exciton resonances (A or B) yields the dynamics proportional to the respective exciton densities in two valleys[14] and provides the valley polarization degree at different delay times and energies. Additional details of the experimental setup are given in Supplementary Note 1.

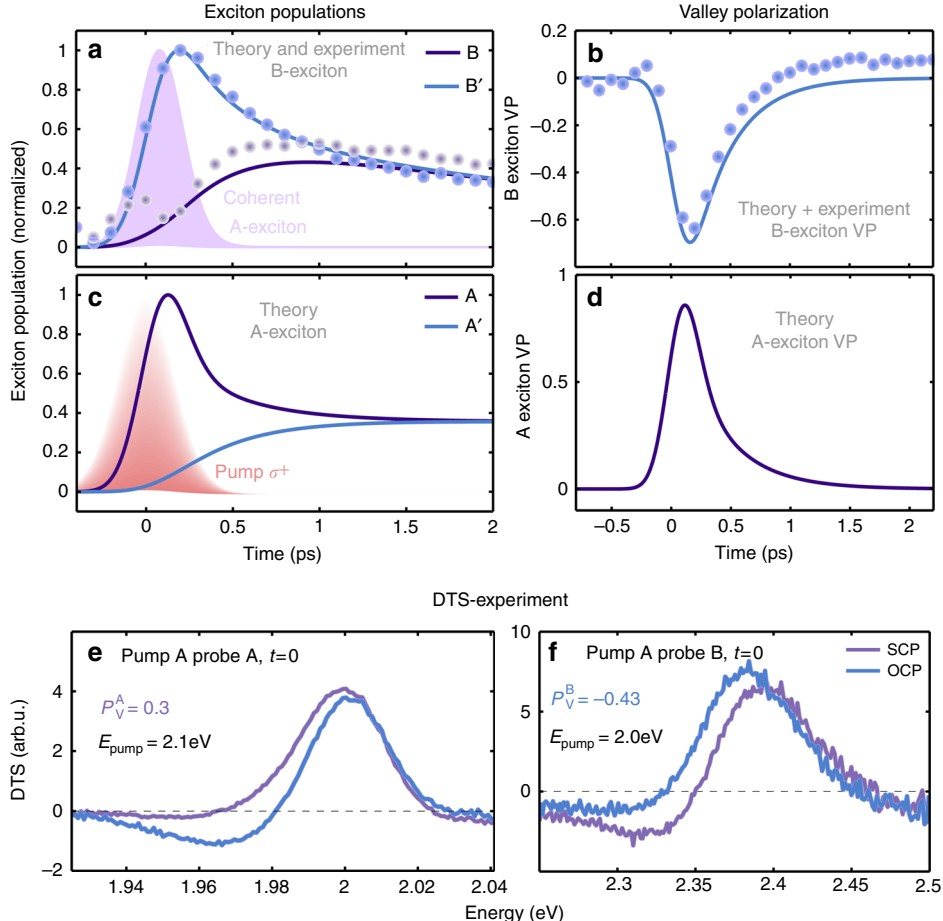

**Fig. 2** Exciton population and valley polarization. Temporal evolution of the exciton population and the resulting valley polarization (VP) of B excitons (**a**, **b** theory–experiment comparison) and A excitons (**c**, **d** theoretical calculation) after resonant excitation of $A_{1s}$ exciton in the K valley. The optically driven coherent A excitons couple via Dexter interaction to the B′ excitons in the unpumped K′ valley (see also Fig. 1). This coupling dominates the B exciton dynamics and instantaneously induces B′ exciton populations (**a**) and as a result a negative valley polarization (**b**) which decays within 0.5 picoseconds. **c** During the pump pulse (red-shaded area), A excitons are generated, which induce A′ excitons via electron-hole exchange interaction. **d** The corresponding optically induced valley polarization of A excitons (near-resonant experimental data in the SI). **e**, **f** Polarization dependent pump-probe measurement showing the OCP (opposite circular polarization) and SCP (same circular polarization) signal of the $A_{1s}$ and $B_{1s}$ excitons, for pumping the $A_{1s}$ exciton close to the resonance

**Time-dependent valley polarization**. We first focus on the dynamics of the valley polarization (VP) $P_v^n$ of A ($n = A$) and B excitons ($n = B$). The latter is defined as $P_v^n = (N_n^K - N_n^{K'})/(N_n^K + N_n^{K'})$. Figure 2 illustrates the time-dependent exciton populations in A and B states after resonant excitation of the $A_{1s}$ exciton in the K valley with a temporally broad (200 fs) and spectrally narrow (20 meV) pump pulse. Simultaneously to the excitation of the A exciton (Fig. 2c), we find an increase in the B′ exciton population (Fig. 2a) resulting in an inversion of the VP (Fig. 2b). This VP follows the pump pulse in time, and it decays as soon as the pump pulse has vanished. The ultrafast dynamics of the VP clearly underlines the coherent nature of the observed process. This is consistently described via the coherent Dexter-like oscillation transfer between the valleys (Fig. 2b, d). The resonantly excited A exciton coherence drives the B′ exciton populations via the Dexter-like coupling. This induces an immediate formation of coherent B excitons in the unpumped valley, followed by a dephasing of the coherence via phonons. This dephasing induces incoherent excitons during the first 200 fs in both A and B′ states. The latter dissipate over the whole Brillouin zone, which leads to a decay of the valley polarization for B and A excitons, which has been characterized by a typical scattering constant $\gamma_{K-K'} = 1$ meV[36]. Furthermore, we have

extracted the lifetime of excitons from the experiments resulting in $\gamma_x = 50$ µeV and $\gamma_x = 200$ µeV for A and B excitons, respectively.

We find in both experiment and theory a strong inversion of the VP of B excitons in the range of −80% when the $A_{1s}$ exciton is resonantly excited (Fig. 2b). However, the theoretical calculation predicts a VP of A excitons of 90%. This is qualitatively in line with the experimentally extracted A exciton time series for near-resonant pump pulses shown in Supplementary Note 2 (Supplementary Fig. 3 for A excitons and Supplementary Fig. 4 for B excitons). For A and B excitons, we find that the VP grows during the excitation process and decays rapidly on a sub-picosecond time scale due to dissipation of the excitonic system. This instantaneous response to the pumping field strongly supports the Dexter-like transfer mechanism, which couples the excitonic coherences of A and B excitons in opposite valleys.

Figure 2e, f show the pump-probe spectra for zero delay time, when the $A_{1s}$ exciton is pumped close to the resonance. In Fig. 2e, the $A_{1s}$ excitons in both valleys (K and K′) are probed. While the SCP curve shows a mainly positive signal, the OCP curve has a dispersive shape and is smaller which indicates the creation of a positive valley polarization[14]. This situation is reversed when probing $B_{1s}$ (Fig. 2f), far away from the initial pump resonance at

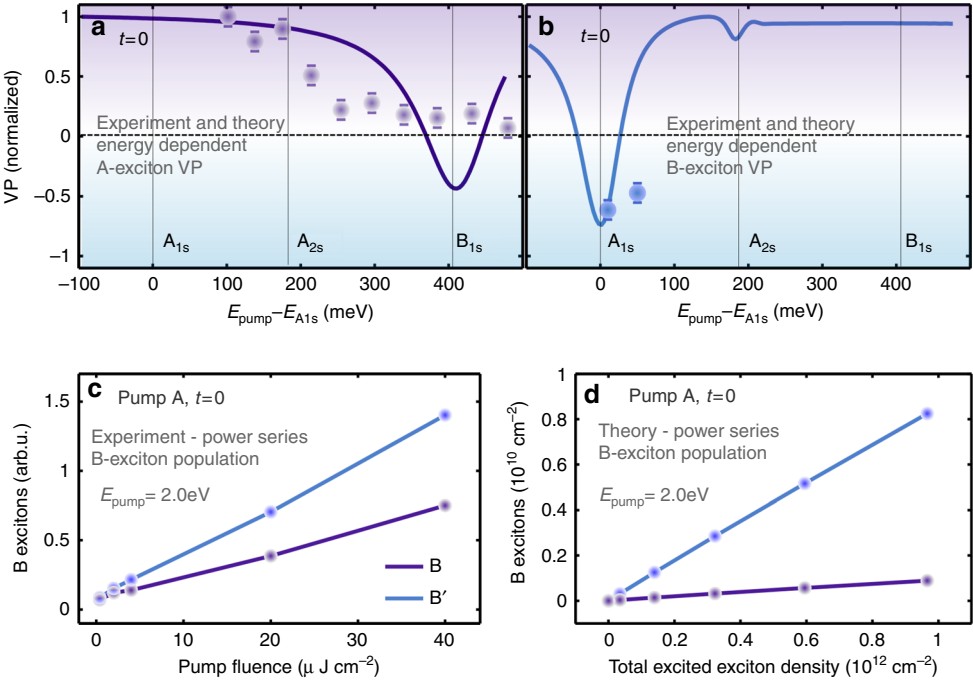

**Fig. 3** Valley polarization as a function of on energy and pump fluence. Valley polarization $P_v^n = \left(N_n^K - N_n^{K'}\right)/\left(N_n^K + N_n^{K'}\right)$ as a function of excitation energy in WS$_2$ for **a** A ($n = A$) and **b** B excitons ($n = B$). Experimental values extracted from pump-probe measurements are shown for both cases as dots. For a better comparison of the qualitative trends, we have normalized the experimental and theoretical values to the maximum measured in experiment and calculated in theory. In the experiment, the A exciton contribution has been fitted by a Voigt function and subtracted from the spectra prior to integration. The limited spectral bandwidth of the measurement leads to uncertainties and determines the width of the error bars of the extracted absolute valley polarization degrees. Due to the Dexter coupling, the A and B valley polarizations are drastically reduced and change sign at excitation energies close to the resonance of the B$_{1s}$ and A$_{1s}$ exciton, respectively. **c, d** Pump fluence dependence of Dexter coupling: Here we show the enhanced excitation of the B exciton as the pump fluence of the excitation pulse (resonant to the A exciton) increases. Due to the larger excitation density we observe in both experiment and in theory an increase of the excitation of B excitons in both valleys. The purple line shows the off-resonantly excited B excitons in the pumped valley, the blue line that B excitons induced via Dexter coupling in the other valley. The latter becomes more efficient for stronger excitation. As a result, we find a higher increase of the population of B excitons in the unpumped valley than in the pumped valley. Note that this is in full analogy to the findings of ref. [19], where an increased signal of the B excitons was observed for larger pump fluences

A$_{1s}$. This shows that the VP for the off-resonant B exciton is inverted with respect to the resonantly pumped A exciton. In addition, in Supplementary Fig. 5 (cf. Supplementary Note 3) we have portrayed the time-dependent polarization of pumped and unpumped valleys. To underline these findings we analyze the population dynamics of the A$_{1s}$ and B$_{1s}$ excitons.

**Pump-dependent valley polarization.** After having revealed the time-resolved exciton population and valley polarization, we now present the impact of Dexter-like coupling when varying the pump energy and the pump fluence at time $t = 0$ fs. First, we analyze the energy dependence of the valley polarization shown in Fig. 3a, b. Here we calculate the generated exciton populations for different pump energies, from 20 meV below A$_{1s}$ up to 50 meV above B$_{1s}$. In the experiments, we tune the excitation energy from 2.10 to 2.48 eV while the pump fluence is fixed at 6 μJ cm$^{-2}$. Figure 3a, b show a direct comparison between the experimental estimated (dots) and theoretical (lines) VP for different pump energies of A and B excitons, respectively. The numerical calculations are in line with analytically estimated intervalley transfer of ~10% for WS$_2$, e.g., the VP is close to +90% for resonant excitation and inverted in the off-resonant case.

Though the theoretical model can reproduce well the trend of the experiment, it does not capture all aspects of the VP dependence for different pump energies. While the theory shows a minor inversion of the A exciton VP, the experiment only indicates a clear reduction, it does not reach the inversion.

In Fig. 3a, both experiment and theory exhibit a pronounced decrease in the VP of A excitons for pump energies in the vicinity of the B$_{1s}$ exciton. We find the inversion only in the case shown in Fig. 3b. Here experiment and theory yield an inverted valley polarization for the B$_{1s}$ excitons for pump energies resonant and close to A$_{1s}$, confirming the crucial role of the Dexter-induced intervalley exciton transfer. Though the theoretical model can reproduce well the trend of the experiment, it does not capture all aspects of the VP dependence for different pump energies. The quantitative discrepancy to the experiment might be due to the fact that the B exciton is in close proximity to a number of higher excitonic A states. This allows additional excitations and coupling processes including phonon-induced scattering and Coulomb exchange coupling. Another difference lies in the experimentally broader absorption spectrum, which can be ascribed to inhomogeneous dephasing and defect-induced absorption, which can reduce the valley polarization.

To reduce the role of defects, we study the resonant excitation of the A exciton while probing the B exciton states for different pump fluences. Under these conditions the major part of the coherence is generated in resonance to the intrinsic states, which reduces the relative amount of defect-assisted absorption. Furthermore, we can control the excitation density in experiment and theory in a comparable way. In Fig. 3c, d, we show the enhanced excitation of the B exciton as the pump fluence of the excitation pulse (resonant to the A exciton) is increased. For larger excitation densities, we observe in both experiment (Fig. 3c) and in theory (Fig. 3d) an increase in the excitation of B excitons

in both valleys. Here the off-resonantly excited B excitons in the pumped valley (purple line) and the B excitons induced via Dexter coupling in the unpumped valley (blue line) are compared. The latter becomes more efficient for stronger excitation, where the phase-space filling induces an asymmetry between the directly excited A exciton and the indirectly (weaker) excited B′ exciton. As a result, the oscillation transfer from the A exciton to B′ is more efficient than the back-coupling process. Therefore, stronger excitation leads to a higher relative increase of indirectly excited B excitons. As a result, we find a higher increase of the population of B excitons in the unpumped valley. Note that this result is in analogy to the findings of ref. [19], where an increased signal of the B excitons was observed for larger CW pump fluences. The inversion of the VP is also evident in the experiment, when pumping the A exciton and probing the B exciton. Here theory and experiment show an inversion of up to −80%.

**Photoluminescence**. To further demonstrate the importance of Dexter-like intervalley coupling, we calculate the valley polarization in the case of continuously pumped WSe$_2$ aiming at the explanation of the recently observed up-conversion in ref. [19]. Here we selectively excite A$_{1s}$ excitons with cw σ$_+$ polarized light. We expect the Dexter coupling to be even more pronounced due to the static excitation leading to ultralong lifetimes of the coupling microscopic polarizations. Evaluating the Bloch equation, we have access to coherent exciton populations $|p_n^{\xi_s}|^2$ that determine the coherent contributions to the photoluminescence[37]

$$I_{\mathrm{PL}}^{\xi_s}(\omega) \propto \mathrm{Im}\left(\sum_n \frac{|p_n^{\xi_s}|^2}{E_n^{\xi_s} - \hbar\omega - i\gamma}\right). \qquad (2)$$

Here we assume a small dephasing of $\gamma = 2$ meV, which is in agreement to the measured excitonic linewidth at low temperatures in ref. [19]. We observe a strong photoluminescence from higher energetic states despite the initial excitation at the A$_{1s}$ exciton corresponding to the experimentally measured up-conversion[19]. In particular, we find a considerable PL signal from the A$_{2s}$ and the B$_{1s}$ state (Fig. 4), which arises from the nonlinear coupling terms in the TMD Bloch equation (second line in Eq. (3) in the Methods section). They enable coupling of excitonic states with different quantum numbers $n,m$ giving rise to an increasing up-conversion to higher excitonic states at higher pump intensities (cf. orange arrows in Fig. 1a). As a result, we find a two orders of magnitude stronger up-conversion from the A$_{1s}$ to the A$_{2s}$ state compared to the case without coupling (cf. dashed lines in Fig. 4). Furthermore, we calculate the PL signal for

σ$_+$ and σ$_−$ polarized light. Without the Dexter coupling (dashed lines), only σ$_+$ is emitted. Switching on the Dexter coupling, we also observe a considerable contribution of σ$_−$ light. The VP for the A$_{1s}$ is reduced to 28%, which is in good agreement with the experimentally estimated value of ~20%[38]. Similarly to the pump-probe experiments, we observe a Dexter-induced inversion of the VP of the B exciton. Here the contribution of the σ$_−$ light is even higher and exceeds the value for σ$_+$ light—again in excellent agreement with the experiment[19].

The Dexter-like intervalley coupling can explain the surprising valley inversion during and shortly after the optical excitation on the sub-picosecond time scale, as well as for CW excitation on longer time scales. For the latter case, it may be an additional process, since one cannot rule out higher-order relaxation mechanisms occurring on a picosecond or a nanosecond time scale. An alternative explanation for the observed up-conversion and inversion of the VP is given in ref. [19], where the inversion is attributed to two-photon absorption followed by a relaxation mechanism under the influence of attractive boson interactions. While this process may be relevant in the CW experiment situation, it can be ruled out in the presented ultrafast pump-probe experiment, where the formation of excitons and the formation of the inverted VP take place on the same time scale (<200 fs)[22,39,40] (cf. Fig. 2).

## Discussion

In summary, we have presented a joint theory–experiment study on the decay of optically accessible valley polarization in TMD. On the basis of a microscopic approach and high-resolution two-color pump-probe experiments, we reveal the crucial importance of the Coulomb-induced Dexter-like intervalley coupling mechanism. We find an efficient coupling of A and B excitonic states in different valleys giving rise to an intervalley transfer of coherent exciton populations. As a result, the valley polarization breaks down and is quasi-instantaneously inverted for B excitons when the A excitonic state is resonantly excited, and vice versa. We show that the upper limit for the achievable valley polarization is given by the ratio of the Dexter coupling strength and the energy difference between A and B excitons that is determined by the spin–orbit coupling. The theoretical prediction is confirmed in spectrally and valley-resolved pump-probe experiments.

## Methods

**Theoretical formalism**. On the basis of the density matrix formalism, we derive the TMD Bloch equation for the microscopic polarization $p^{ij}$ within the coherent limit[20,21,41]. To account for the crucial importance of excitons, we project the solution for the microscopic polarization into an excitonic basis using the transformation $p_{\mathbf{k}}^{vc\xi_s} = \sum_n p_n^{\xi_s} \theta_{\mathbf{k}}^{n\xi_s}$. The appearing excitonic wave functions $\theta_{\mathbf{k}}^{n\xi_s}$ are calculated by solving the Wannier equation, an eigenvalue equation for excitons that also provides the excitonic energies $E^{n\xi_s}$ of the state $n$[20,21,31]. The TMD Bloch equation reads in the excitonic basis for the microscopic polarization:

$$i\hbar\dot{p}_n^{\xi_s} = \tilde{E}_n^{\xi_s} p_n^{\xi_s} - \sum_{\mathbf{k}}\left(\Omega^{n\xi_s} - \sum_{ij}\tilde{\Omega}_{ij}^{n\xi_s}\left(p_i^{\xi_s}p_j^{\xi_s*} + \Delta N_{ij}\right)\right)$$
$$+ \sum_{m,i,j}\left(\Gamma_{ij}^{nm\xi_s\xi_s}\left(p_i^{\xi_s}p_j^{\xi_s*} + \Delta N_{ij}\right)\right)p_m^{\xi_s}$$
$$- \sum_{m,i,j}\left(Y_{ij}^{nm\xi_s\bar{\xi}_s}\left(p_i^{\xi_s}p_j^{\bar{\xi}_s*} + \Delta N_{ij}\right)\right)p_m^{\xi_s}$$
$$- \sum_m\left(D^{nm\xi_s\bar{\xi}_s} - \sum_{i,j}\tilde{D}_{ij}^{nm\xi_s\bar{\xi}_s}\left(p_i^{\xi_s}p_j^{\xi_s*} + \Delta N_{ij}\right)\right)p_m^{\bar{\xi}_s}. \qquad (3)$$

inducing incoherent exciton densities via phonon scattering,

$$i\hbar\Delta\dot{N}_{ij}^{\xi_s} = 2\gamma_{ph}p_i^{\xi_s}p_j^{\xi_s*} - \gamma_x\Delta N_{ij}^{\xi_s} - \gamma_{\xi,\bar{\xi}_s}\Delta N_{ij}^{\bar{\xi}_s} \qquad (4)$$

The dynamics of the excitonic polarization $p_n^{\xi_s}$ is driven by an external optical field denoted by the Rabi frequency $\Omega^{n\xi_s}(t) = \sum_{\mathbf{k}} \theta_{\mathbf{k}}^{n\xi_s*}\mathbf{M}_{\mathbf{k}}^{vc\xi_s}\cdot\mathbf{A}(t)$ including the

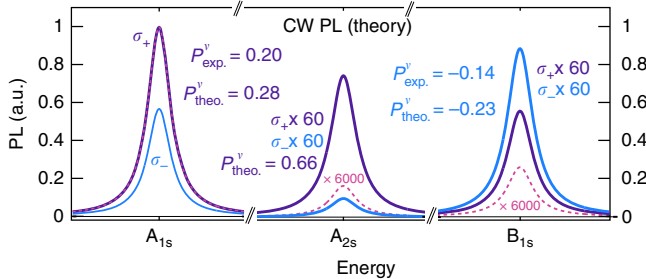

**Fig. 4** CW photoluminescence valley polarization for A and B excitons. Theoretical cw photoluminescence in WSe$_2$ showing the emitted σ$_+$ and σ$_−$ polarized light at A$_{1s}$, A$_{2s}$, and B$_{1s}$ states. The difference between the σ$_+$ and the σ$_−$ contribution directly reflects the valley polarization of the corresponding states. The thin dashed lines show the PL without the Dexter coupling. The obtained values for the valley polarization are compared to the values measured in recently performed cw PL experiments in ref. [19]

vector potential $\mathbf{A}(t)$ and the optical matrix element $M_{\mathbf{k}}^{vc\xi_s}$ weighted by the excitonic wave function $\theta_{\mathbf{k}}^{n\xi_{s^*}}$. The optical matrix element is analytically obtained by using nearest-neighbor tight-binding wave functions[31], Non-linear phase-space filling effects have also been taken into account in $\tilde{\Omega}_{ij}^{n\xi_s}(t) = \sum_{\mathbf{k}} 2\theta_{\mathbf{k}}^{i\xi_s}\theta_{\mathbf{k}}^{j\xi_{s^*}}\Omega_{\mathbf{k}}^{n\xi_s}(t)$. Furthermore, the excitonic polarization oscillates with $\tilde{E}_n^{\xi_s} = (E_n^{\xi_s} + i\gamma_{hom})$, where $\gamma_{hom}$ corresponds to the radiative decay rate determining the homogeneous dephasing of the polarization[16,42] (cf. Supplementary Fig. 2).

The second and the third line in Eq. (3) describe the Coulomb-induced inter-excitonic coupling ($\Gamma$ term) and intervalley bandgap renormalization (Y-term). The first couples states of the same spin within one valley (Fig. 1), while the latter leads to a red-shift of excitonic resonances in one valley due to the optically excited carrier occupations in the other valley[14]. The corresponding coupling elements read

$$\Gamma_{ij}^{nm\xi_s,\xi_s} = \sum_{\mathbf{kk'}} \theta_{\mathbf{k}}^{n\xi_{s^*}} \left( 2\theta_{\mathbf{k}}^{i\xi_s}\theta_{\mathbf{k'}}^{j\xi_s}\theta_{\mathbf{k'}}^{m\xi_s} V_{c\xi_s v\xi_s,\mathbf{k'k}}^{c\xi_s v\xi_s,\mathbf{kk'}} \right.$$
$$\left. -\theta_{\mathbf{k'}}^{i\xi_s}\theta_{\mathbf{k'}}^{j\xi_s}\theta_{\mathbf{k}}^{m\xi_s} \sum_{\lambda} V_{\lambda\xi_s\lambda\xi_s,\mathbf{k'k}}^{\lambda\xi_s\lambda\xi_s,\mathbf{kk'}} \right), \qquad (5)$$

$$Y_{ij\mathbf{kk'}}^{nm\xi_s,\bar{\xi}_s} = \theta_{\mathbf{k}}^{n\xi_{s^*}}\theta_{\mathbf{k}}^{i\bar{\xi}_s}\theta_{\mathbf{k'}}^{j\bar{\xi}_{s^*}}\theta_{\mathbf{k}}^{m\xi_s} \sum_{\lambda} V_{\lambda\bar{\xi}_s\lambda\xi_s,\mathbf{k'k}}^{\lambda\xi_s\lambda\bar{\xi}_s,\mathbf{kk'}} \qquad (6)$$

with the screened Coulomb matrix elements $V_{\lambda\bar{\xi}_s\lambda'\xi_s,\mathbf{k'k}}^{\lambda\xi_s\lambda'\bar{\xi}_s,\mathbf{kk'}}$.

Finally, the last line in Eq. (3) describes the intrinsic Coulomb-induced intervalley coupling of excitonic states in different valleys and has been discussed in the main text. Similarly to the Rabi frequency we also consider non-linear phase-space filling effects scaling with $\tilde{D}_{ij}^{nm\xi_s,\xi_s} = \sum_{\mathbf{kk'}} 2\theta_{\mathbf{k}}^{i\xi_s}\theta_{\mathbf{k}}^{j\xi_{s^*}} D^{nm\xi_s,\xi_s}$.

The appearing Coulomb matrix elements are calculated within the nearest-neighbor tight-binding approach and by exploiting an effective Keldysh screening $V_q = V_q^{2D}\varepsilon_q^{-1}$ which has been demonstrated to describe excitonic properties well in atomically thin two-dimensional materials on a specific substrate[28,31,43]. Note, however, that the regular Keldysh screening with $\varepsilon_q = 1 + r_0 q$ can overestimate the screenings for processes with a large momentum transfer. Since Dexter coupling bridges different valleys, we have defined an effective Keldysh screening $\varepsilon_q = 1 + r_0 q/((qa_0)^{5/2} + 1)$ that has been adjusted to recent DFT calculations[35]. Here $a_0 \approx 0.3$ nm denotes the lattice constant, $r_0 = \varepsilon_\perp d/(\varepsilon_s)$ is the screening length with the dielectric tensor of the monolayer $\varepsilon_\perp \approx 11.7$[28] and the surrounding substrate $\varepsilon_s = \varepsilon_1 + \varepsilon_2$, and the thickness of the material is approximated to 0.7 nm, matching the experimentally measured spectral separation between A$_{1s}$ and A$_{2s}$ states. The general form of the screened Coulomb matrix elements reads:

$$V_{\lambda\xi_s\lambda'\xi_s,\mathbf{k'k}}^{\lambda\xi_s\lambda'\bar{\xi}_s,\mathbf{kk'}} = \frac{C_{\mathbf{k,k'}}^{\lambda\lambda,\xi\bar{\xi}} C_{\mathbf{k',k}}^{\lambda'\lambda',\bar{\xi}\xi} V_{\mathbf{k-k'}+\Delta_{\xi\bar{\xi}}}^{2D}}{\varepsilon_{\mathbf{k-k'}+\Delta_{\xi\bar{\xi}}}} \qquad (7)$$

where $C_{\mathbf{k,k'}}^{ii,lj}$ are the valley-dependent phase factors determined by lattice symmetry[34]. Furthermore, the transferred momentum $q = \left| \mathbf{k} - \mathbf{k'} + \Delta_{\xi\bar{\xi}} \right|$ corresponds to the distance of the involved states located within one valley ($\Delta_{\xi\xi} = 0$) or in different valleys ($\Delta_{\xi\bar{\xi}} = \mathbf{K} - \mathbf{K'}$).

The values of the effective masses $m_s^\lambda$ of the spin-degenerated parabolic bands around the K points are taken from PBE-based DFT calculations[44]. In WS$_2$ they correspond to $m_\uparrow^c = 0.36$, $m_\downarrow^c = 0.27$ for the conduction band, and $m_\uparrow^v = 0.36$, $m_\downarrow^v = 0.50$ for the valence band.

**Data availability**. The data that support the findings of this study are available from the corresponding author upon request.

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

## Acknowledgements

This project has received funding from the European Unions Horizon 2020 research and innovation program under grant agreement No 696656—Graphene Flagship (E.M.). Furthermore, we acknowledge support by Stiftelsen Olle Engkvist (I.B.-V.), Swedish Research Council (G.B. and E.M.) as well Deutsche Forschungsgemeinschaft through the SONAR EU Project (A.K.) and SFB 787 (G.B. and A.K.).

## Author contributions

G.B. and I.B.-V. have performed the theoretical calculations of the valley polarization. The experimental studies have been carried out by the coauthors from the University of Münster. All coauthors have contributed to the writing of the manuscript and the interpretation of results.

## Additional information

**Competing interests:** The authors declare no competing interests.

