## [Peer Review File · Nature Communications]

Reviewers' comments:

Reviewer #1 (Remarks to the Author):

In their present work "Inverted valley polarization in optically excited transition metal dichalcogenides" by Berghäuser et al, the authors propose a Dexter-like intervalley coupling mechanism that couples A and B excitons in adjacent valleys in transition metal dichalcogenide monolayers (TMDC).

Their theoretical modeling is accompanied by optical spectroscopy measurements in WS₂ monolayers that reveal a clear inversion of valley polarization of B excitons when A excitons are resonantly excited, and a drop of valley polarization of A excitons when B excitons are resonantly photogenerated.

While their work is of considerable interest for researchers in the field of multi-valley 2D materials, I don't think that the presented experimental results are novel enough to be considered for publication in Nature communications. Indeed, their experimental data has already been reported by different groups in WS₂ and other related monolayer materials of the TMDC family (Scientific Reports 6, Article number: 18885 (2016), Applied Physics Letters 106 (11), [112101], and reference [18] cited in the article). Moreover, their claim about other models being not consistent with their observations is difficult to justify since it is not possible from their data to discriminate between their scenario and other possible effects (see below).

Please find below a detailed list of issues and suggestions:

1-The authors argue that the observed up-conversion and inversion of valley polarization cannot be explained by a second-order process followed by energy relaxation as proposed in Ref.[18]. Their main argument is that the observed inversion of polarization is ultrafast and occurs within the resolution of the pump-probe experiment (zero delay). It is not clear, however, why another scenario such as the one proposed in Ref.[18] should be slower than the Dexter-like intervalley coupling. Indeed, several time-resolved experiments have shown an ultrafast exciton formation (Nano Lett., 2017, 17 (3), pp 1455–1460) even for far-above resonant excitation and sub-picosecond energy relaxation (Nano Lett., 2017, 17 (7), pp 4210–4216) in TMDC monolayers.

In addition, as shown in Ref.[18], the inversion of valley polarization depends crucially on the exciton density, whereas the model proposed by the authors is unable to explain this dependence and predicts an inversion of valley polarization regardless of the pump fluence.

2.For the up-conversion experiment, only one pump fluence was used and one data point is presented. A study including the power-dependence and pump energy is desirable in order to better confront experiment and theory.

3- It is not clear how to discriminate experimentally between the proposed Dexter-like coupling between A and B excitons and other possible mechanisms, for example the one in which an A-exciton with non-zero kinetic energy in one valley couples to a B exciton in the adjacent valley via the exchange interaction. Another possibility is 2-photon absorption since this process will couple, for example, two right-circularly polarized photons to states which couple to left-circularly polarized photons. This could also result in a smaller valley polarization when the excitation energy is resonant with the B exciton since the latter has a large oscillator strength.

Reviewer #2 (Remarks to the Author):

This manuscript by Berghäuser et al. discusses intervalley coupling in transition metal dichalcogenides through both theory and experiment. The claim is that there is a coherent

coupling between A and B excitons that can lead to oscillations between the excitons. The paper is generally well-written, and adds new insights to the field of intervalley coupling in TMDs. The theoretical discussion using semiconductor Bloch equations is generally a clear description of the author's ideas, and straightforward to follow. The author's approach with Dexter-like interaction adds to the literature on intervalley coupling mechanisms, and could prompt new experiments. The experimental part includes comparisons to existing literature and additional data. I find the experimental data included in this manuscript to be less definitive, but a nice addition to the theoretical discussion that improves the manuscript in its entirety.

In summary, I find that this paper is of interest to the field. The theoretical treatment with supporting experimental data creates a compelling picture that will advance the existing literature. I think the manuscript is suitable for publication in Nature Communications, although I do have technical comments and questions that I believe the authors should try to address to improve the manuscript going forward.

- Eq. 1 is referred to on page 2 as the "Semiconductor Bloch Equation". I am not sure what is referred to here, as Eq. 1 is not a Bloch equation but rather a matrix element expression. This language should be clarified.

- Reference is made to data in Ref. 18 on inverted valley polarization. That reference primarily uses 2-photon excitation to the continuum to describe their results. The authors here claim that the explanation of Ref. 18 is not consistent with the ultrafast data here, but the Dexter mechanism can explain all of the experimental data here and in Ref. 18. This is an intriguing claim that extends the impact of these results. Can it be supported by some timescales? The claim that 2nd order absorption and relaxation cannot explain the quasi-instantaneous depolarization would depend on the timescales of the depolarization. What are they? Are they slower than the pump-probe resolution or faster?

- The key results/predictions for ultrafast and CW experiments are in Fig. 3a,b. Two main features are reported: a decrease of A exciton polarization with higher pump energy, which would be expected, and a single point showing inverted B exciton polarization with A exciton energy pumping. The experimental aspects of these features could be investigated more completely. Regarding pump energy decrease of the A exciton polarization, it is expected from previous measurements. Is there any feature specific in the data here that suggests that the Dexter model is correct? The most noticeable feature (inverted polarization at >2.3 eV) is not observed. Are other explanations possible for the data in 3a?

- Fig. 3a,b presents experimental data on top of theory curves. From the author's discussion, the "key" observation supporting the claim of Dexter-like coupling would be the inversion of polarization of the oppositely pumped excitons. There seems to be a single data point in Fig. 3b supporting this claim, and its error bar ranges over -25% to -75% for the B exciton when the A exciton is pumped. This single data point is not very convincing...is the error bar a 66%, 95%, 99% confidence interval? The caption suggests that the error bar is extreme values of the measurement, and not the uncertainty? If so, is it possible to adjust the model to account for the spectral dependence (Fig 3c,d) so that the theory and data are comparing the same thing and such a non-standard error bar (which does not represent error but spectral range) is not needed?

- The interesting inverted point in Fig. 3b should be expanded on with additional points in the curve (specifically...does the inverted polarization disappear as the pump energy is increased for 2.05 eV?) This would highlight if the predicted *trend* is observed, rather than just a single point.

- The discussion of Fig. 3e in the text was confusing. The discussion in p. 5 below Eq. 2 explicitly states that calculations are being discussed, but at parts implies that experimental data is being analyzed, using words like "excite" and "observe". If there are indeed experimental measurements

(beyond the upconversion referenced in Ref. 18), these must be better highlighted and presented. If there are not experiments for the CW PL, then the discussion following Eq. 2 should make this clear. The comparison to Ref. 18 data is good, but why is there no observed polarization of A2s reported in this experiment? Additional CW PL data to support the entire picture of Fig. 3e would be more convincing. Is this CW experiment available to test the entire prediction of CW PL?

- The Methods provide an outline of the calculations performed, but it is not overly detailed on the procedure. The authors may consider including more numerical details in the Supplementary on how wavefunctions and coupling parameters are calculated and what numbers are used as inputs, which would aid someone in reproducing these calculations.

Reviewer #3 (Remarks to the Author):

In this manuscript, the authors have presented a joint theory-experiment study to show the decay of spin and valley polarization in transition metal dichalcogenides (TMDs). This present work reveals a new Coulomb driven intervalley coupling mechanism, that determines the optically accessible spin valley polarization in TMDs. The interaction resembles Dexter coupling between K and K' valleys.

In general, these results add some new information about monolayer TMDs and could be of interest to specialists in the field, however I doubt that the presented results are novel enough to be published in Nature Communications. I recommend redirecting this manuscript to another more specialized journal.

In addition, authors should consider to address the following issues before resubmitting to any other journal:

1) The authors should specifically mention if figure 2 represents experimental data or theoretical modelling. I assume this is theoretical modelling.

Further it's unclear if the data demonstrates a pump-probe based measurement or a time resolved PL polarization intensity (at zero delay).

Moreover the polarization should be maximum at time = 0 fs. Why the peak is shifted from zero?

2) The authors assume the homogeneous width of the "B" exciton = 50 meV (PAGE 3).

I believe they found this number assuming homogeneous width of "A" exciton = 10 meV [from Ref. 15]. Since the absorption data in the supplemental material exhibits the linewidth of the "B" exciton feature is ~5 times broader than the "A" exciton linewidth, thus homogeneous width of "B" exciton = 50 meV. This approach is incorrect.

The linewidth obtained from the absorption is usually governed by inhomogeneous distribution of exciton frequencies. Even at low temperature, the intrinsic homogeneous width is masked by the inhomogeneity. One needs to employ either Four wave mixing [e.g. PRL 116, 127402 (2016)] or two dimensional Fourier transform spectroscopy [e.g. Nature Communication, 6, 8315, 2015] to estimate the homogeneous width correctly in TMDs.

3) Figure 3b exhibits only one experimental data point against the theoretical modelling. Authors need to show more experimental data.

4) The authors should explain their experimental details more elaborately. For example: In DTS

measurements, what is the pump pulse width? Where the (250 meV) probe pulse is peaked at? Details regarding laser sources (for both pump and probe), and their repetition rates are also need to be mentioned.

5) I fail to understand the purpose of introducing the case of MoS₂ (PAGE 3) where the authors predict 25% indirect absorption. This diverts the focus as the authors have neither exhibited any experimental or theoretical data from MoS₂.

Some minor points:

a) The dielectric constant (ϵ) for SiO₂ is generally varies between 3.9 and 4.0. Is there any specific reason that the authors used $\epsilon = 2.1$ for their calculation (in order to estimate the coupling strength in PAGE 2)?

b) In equation 1, what are the coefficients (Γ) signifies? Please explain.

Reviewer 1

1. Comment:

While their work is of considerable interest for researchers in the field of multi-valley 2D materials, I don't think that the presented experimental results are novel enough to be considered for publication in *Nature communications*. Indeed, their experimental data has already been reported by different groups in WS2 and other related monolayer materials of the TMDC family (*Scientific Reports* 6, Article number: 18885 (2016), *Applied Physics Letters* 106 (11), [112101], and reference [18] cited in the article). Moreover, their claim about other models being not consistent with their observations is difficult to justify since it is not possible from their data to discriminate between their scenario and other possible effects (see below).

Response:

We thank the reviewer for carefully reading our manuscript and for the generally positive evaluation of our work. We have performed new experiments and calculations which underline the presented findings, their novelty, and importance. In particular, we have measured and simulated the temporal evolution of the valley polarizations showing a quasi-instantaneous excitation-induced increase followed by a rapid decay within 500 fs, cf. Fig.1. The instantaneous response is a strong support of the Dexter-transfer, which shows no time delay. We further demonstrate that the valley polarization increases with the applied pump fluence, where both experiment and theory show an excellent agreement, cf. Fig. 2.

Finally, we do not understand the statement that the presented findings have been already reported in previous publications. The articles mentioned by the referee report on photoluminescence studies and hence do not contain any information on the sub-picosecond time scale (Fig.1), which is exactly the focus of our work. We point out for the first time the importance of Dexter-like intervalley coupling, which can explain the surprising valley inversion during and shortly after optical excitation on the sub-picosecond time scale as well as for continuous wave (CW) excitation on longer time scales. At the same time, for the CW experiments we cannot rule out higher-order relaxation mechanisms occurring on a ps or ns time scale. However, to exploit the potential of TMDs for spin- and valleytronics devices the optical excitation has to be ultrafast to compete with current technology. Hence, the newly discovered instantaneous Dexter-like coupling mechanism is of utmost importance. Therefore, our results are new and important for the entire community of 2D materials and related heterostructures. The Dexter-like coupling mechanism could even be relevant for other highly correlated quantum systems.

We have considerably modified the resubmitted manuscript to address the mentioned points of the referee. In particular, we have included Figs. 1 and 2 from the reply to the main manuscript.

FIG. 1: Time-dependent B-exciton valley polarization: The pump pulse is resonant to the A_{1s} exciton and has a duration of 250 fs. Simultaneously to the excitation of the A exciton we find an increase of the inverted valley polarization of the B exciton. The inversion of the valley polarization follows the pump pulse in time, and in analogy to the direct optical excitation it decays as soon as the pump pulse has vanished. This clearly underlines the coherent nature of the observed process.

FIG. 2: **Pump fluence dependence of Dexter coupling:** Enhanced excitation of the B exciton is observed both in experiment and theory as the pump fluence of the excitation pulse (resonant to the A exciton) increases. The purple line shows the off-resonantly excited B excitons in the pumped valley (for the theoretical calculations the line has been shifted by 4.7% for a better comparison), while the blue line shows the B excitons induced via Dexter coupling in the unpumped valley. The latter becomes more efficient for stronger excitation resulting in a higher increase of the B exciton population compared to the pumped valley. Note that this is in analogy to the findings of Ref. 18, where an increased signal of B excitons was observed for larger pump fluences.

2. Comment:

The authors argue that the observed up-conversion and inversion of valley polarization cannot be explained by a second-order process followed by energy relaxation as proposed in Ref.[18]. Their main argument is that the observed inversion of polarization is ultrafast and occurs within the resolution of the pump-probe experiment (zero delay). It is not clear, however, why another scenario such as the one proposed in Ref.[18] should be slower than the Dexter-like intervalley coupling. Indeed, several time-resolved experiments have shown an ultrafast exciton formation (Nano Lett., 2017, 17 (3), pp 14551460) even for far-above resonant excitation and sub-picosecond energy relaxation (Nano Lett., 2017, 17 (7), pp 42104216) in TMDC monolayers.

Response:

We thank the reviewer for bringing up this important point. We agree that the formation time of excitons occurs on the sub-picosecond time scale. This is found not only theoretically (arXiv:1703.03317), but also experiments by Steinleitner et al (Nano Lett., 2017, 17 (3), pp 14551460) underline this result. However, after a two-photon absorption and hot exciton formation at 4 eV [18] the excitons would have to relax down (cooling) to 2.4 eV where the B exciton is located. Assuming a typical phonon energy of 40 meV, the direct relaxation path consists of a cascade emission of 40 phonons, which is unlikely to occur on a sub-picosecond timescale. E.J. Sie et al (Nano Lett., 2017, 17, 4210) estimate a cooling time of 2 ps for above band-edge excitation. However, this estimation is based on energetic shifts of the A-exciton resonance after the pump pulse. The shifts are explained as a consequence of different screening of the Coulomb potential implied by free carriers and excitons. The latter induce energetic shifts of all states (occupied and unoccupied) in the semiconductor. There are no theoretical studies of the relaxation mechanism or experimental evidences proving an occupation of A_{1s} or B excitons on the sub ps time scale after excitation. Furthermore, a cooling time of 2 ps implies an exciton-phonon scattering time of less than 50 fs and a huge homogeneous broadening of at least 150 meV yielding a line width of 300 meV for every exciton state involved in the cascade process. This clearly contradicts experimental observations (including reference [18]) showing that the excitonic linewidths also of higher excitonic states are rather in the range of a few tens of meV (Solid State Communications 203, 2015, 16 + Nature Communications 6, 8315 (2015) + 2D Mater. 4, 031011 (2017)).

In the revised work, we have performed new experiments investigating the temporal evolution of the valley polarization of the B exciton. In Fig. 1, the experimentally measured time-dependent valley polarization of B excitons is shown. We find that the inversion of the valley polarization increases during the presence of the pulse and decreases after the pump pulse has vanished (around 0.2 ps). If there were two-photon absorption in this experiment, followed by ultrafast relaxation inducing an inverted valley polarization, it should show a delayed increase during the cooling time of 2ps, which is not the case. We have extended the discussion on this crucial point in the main manuscript and have further added a paragraph on the new experimental findings on the time-dependent valley polarization of the B exciton.

3. Comment:

In addition, as shown in Ref.[18], the inversion of valley polarization depends crucially on the exciton density, whereas the

model proposed by the authors is unable to explain this dependence and predicts an inversion of valley polarization regardless of the pump fluence.

Response:

The referee addresses an important point. We have performed new experiments on the pump fluence dependence showing the same trend as in Ref. [18], cf. Fig.2 (a). Furthermore, we want to emphasize that our equations do include density-dependent effects. In particular, the phase space filling, which becomes important for stronger excitations, induces an asymmetry between the directly excited A excitons and the indirectly (weaker) excited B excitons (Eq. 3 of the methods section). As a result, strong excitation leads to a higher relative increase of indirectly excited B excitons, as shown in Fig. 2 (b). Our theory, which includes density dependent phase space filling effects, is in excellent agreement with the experimental measurements showing an increase of the indirectly driven B' excitons with increasing pump fluence. Note however that in the high excitation regime also non-linear exciton-exciton and Auger scattering terms become important. In a full microscopic study all these processes would need to be included, which is beyond the scope of the present work.

We have added a new paragraph in the main manuscript including the new experimental data on the pump fluence dependence of the valley polarization.

4. Comment:

For the up-conversion experiment, only one pump fluence was used and one data point is presented. A study including the power-dependence and pump energy is desirable in order to better confront experiment and theory.

Response

We agree with the reviewer that it would be preferable to measure a complete energy series for the B exciton similar to the series we measured for the A exciton. Unfortunately, in our current collinear experimental setup it is not possible to perform a degenerate pump-probe measurement, where the pump laser energy is in the probe energy range. Since the B exciton shifts towards lower energy due to band gap renormalization when the A exciton is pumped, we have to probe the energy range from 2.07 eV to 2.58 eV to obtain an evaluable signal. However, we have managed to get at least one additional data point, where the monolayer is pumped at 2.05 eV. This new data is shown in the supporting information (Figure S5(c) and (d)). In this measurement, the pump laser light is already leaking through the long-pass filter into the pump-probe signal, which had to be removed in the data processing. Therefore, the extracted polarization degree has a larger uncertainty and a comparison with the resonant case (pump at 2.01 eV) has to be done very carefully. Nevertheless the inverted valley polarization can be seen clearly and the absolute value is -0.33. Additionally, we have performed a power series of the B exciton valley polarization for the case of a resonantly pumped A exciton, as it was suggested by the reviewer. The results are discussed in the response to comment 3 and have been added to the manuscript.

5. Comment:

It is not clear how to discriminate experimentally between the proposed Dexter-like coupling between A and B excitons and other possible mechanisms, for example the one in which an A-exciton with non-zero kinetic energy in one valley couples to a B exciton in the adjacent valley via the exchange interaction. Another possibility is 2-photon absorption since this process will couple, for example, two right-circularly polarized photons to states which couple to left-circularly polarized photons. This could also result in a smaller valley polarization when the excitation energy is resonant with the B exciton since the latter has a large oscillator strength.

Response

We thank the reviewer for addressing again a very important aspect. We fully agree that there are also other possible scattering processes and we cannot rule out their existence. However, in the presented sub-picosecond study we are convinced that these effects are negligible for the following reason: K-phonons have an energy of 40 meV. In the proposed scenario, the A exciton would have to have a kinetic energy of about 360 meV to overcome the spin-orbit coupling of 400 meV. First, the up-scattering of A excitons from the light cone to the states 360 meV above requires the absorption of 9 phonons, which is unlikely to occur within 200 fs. Second, assuming that the thermalization into an equilibrium Bose-like distribution via phonons appears quasi-instantaneously or that maybe disorder dissipation would lead to A-excitons with sufficient non-zero kinetic energy could also not explain the strong signals found in the experiment, since the exchange interaction mentioned by the referee is a momentum-conserving Coulomb process. The latter implies that an A-exciton with non-zero kinetic energy would only couple to B-excitons also with non-zero kinetic energy. This means that the coupling process is still extremely off-resonant and highly inefficient. As a result, although these processes are in principle possible, they are expected to be small compared to the discussed coupling processes in the investigated case.

We have extended the discussion on alternative mechanisms in the revised version of the manuscript.

Reviewer 2

1. Comment

In summary, I find that this paper is of interest to the field. The theoretical treatment with supporting experimental data creates a compelling picture that will advance the existing literature. I think the manuscript is suitable for publication in Nature Communications, although I do have technical comments and questions that I believe the authors should try to address to improve the manuscript going forward.

Response

We thank the referee for carefully reading our manuscript and for her/his clear recommendation of our work for publication in Nature Communications.

2. Comment

Eq. 1 is referred to on page 2 as the Semiconductor Bloch Equation. I am not sure what is referred to here, as Eq. 1 is not a Bloch equation but rather a matrix element expression. This language should be clarified.

Response

We thank the referee for drawing our attention to this point. We have corrected this misprint in the new version of our manuscript and have additionally extended the theoretical discussion in the supplementary material.

3. Comment

Reference is made to data in Ref. 18 on inverted valley polarization. That reference primarily uses 2-photon excitation to the continuum to describe their results. The authors here claim that the explanation of Ref. 18 is not consistent with the ultrafast data here, but the Dexter mechanism can explain all of the experimental data here and in Ref. 18. This is an intriguing claim that extends the impact of these results. Can it be supported by some timescales? The claim that 2nd order absorption and relaxation cannot explain the quasi-instantaneous depolarization would depend on the timescales of the depolarization. What are they? Are they slower than the pump-probe resolution or faster?

Response

We thank the referee for bringing up this very important point. We have performed new experiments and calculations and have added a corresponding paragraph to the revised manuscript (see also the response to the 2. comment of the first referee). The cooling of hot excitons after two-photon excitation should not appear instantaneously and should lead to a delayed signal. We have recorded a time series of the pump probe experiment showing the dynamics of the valley polarization, cf Fig.1. We observe that the valley polarization follows the pump pulse and instantaneously decays as soon as the pulse has vanished. This shows that the presented effect is directly linked to the excitation process. We do not observe a time delay of the signal due to hot carrier relaxation into the B_{1s} state as would be expected after two-photon excitation. Therefore, we are convinced that in the presented pump-probe study the two-photon absorption does not play a significant role. We also find that our theory can indeed explain both results, the DTS signal on the sub-picosecond time scale and the CW PL signals at long time scales. However, in the latter case the theory can be seen as an alternative or additional process which does not contradict the results or explanation of Ref. 18.

We have extended the discussion on this point in the revised manuscript and have clarified our theoretical predictions with respect to other possible processes.

4. Comment

The key results/predictions for ultrafast and CW experiments are in Fig.3a,b. Two main features are reported: a decrease of A exciton polarization with higher pump energy, which would be expected, and a single point showing inverted B exciton polarization with A exciton energy pumping. The experimental aspects of these features could be investigated more completely. Regarding pump energy decrease of the A exciton polarization, it is expected from previous measurements. Is there any feature specific in the data here that suggests that the Dexter model is correct? The most noticeable feature (inverted polarization at ≈ 2.3 eV) is not observed. Are other explanations possible for the data in 3a?

Response

We thank the reviewer for her/his valuable suggestion. We have performed new experiments and calculations, which further underline the importance of the Dexter coupling:

- We have recorded the population dynamics (Fig.1) showing that the inversion of the valley polarization follows a typical profile of optical excitation. This clearly indicates that the underlying process is directly linked to the excitation and not to a delayed relaxation process.
- We have performed a power series to investigate the dependence of the coupling on the excitation strength (Fig.2). We find in both experiment and theory an increase of the valley polarization inversion in WS₂. Note that this is also in line with CW measurements of Ref. 18. While the quantitative trend agrees remarkably well in experiment and theory, Fig. 2 also shows the limits of theory-experiment comparison. The theoretically achievable valley polarization clearly exceeds the one in the experiment. This can be ascribed to the fact that the valley polarization in experiment is strongly influenced by a reduced circular dichroism and by defects in real TMD monolayers.

We have included discussion in the revised manuscript clarifying the above points.

5. Comment

Fig. 3a,b presents experimental data on top of theory curves. From the authors discussion, the key observation supporting the claim of Dexter-like coupling would be the inversion of polarization of the oppositely pumped excitons. There seems to be a single data point in Fig. 3b supporting this claim, and its error bar ranges over -25% to -75% for the B exciton when the A exciton is pumped. This single data point is not very convincing is the error bar a 66%, 95%, 99% confidence interval? The caption suggests that the error bar is extreme values of the measurement, and not the uncertainty? If so, is it possible to adjust the model to account for the spectral dependence (Fig 3c,d) so that the theory and data are comparing the same thing and such a non-standard error bar (which does not represent error but spectral range) is not needed?

Response

We agree that the large error bar might be confusing and want to explain its origin. Actually, the large error bar is not a result from the measurement uncertainties, but due to the analysis of the measured data. To extract the B exciton populations in both valleys from the pump-probe data, we spectrally integrate the signature of the B exciton in the pump-probe data. Doing this eliminates the effects of spectral broadening and shifting of the exciton resonance from the pump-probe signal, yielding a signal which depends only on the bleaching of the resonance, which corresponds to the exciton population. It is very important to only integrate over the B exciton signature. The background from overlapping features from the A exciton or excited states of the A exciton has to be removed from the signal to extract the correct population of the B exciton. Indeed, integrating with background or removing it before has a large influence on the derived valley polarization degree. Hence, the limits of the error bar result from the two extreme cases: completely neglecting the background or fully removing it. We have now performed new measurements (power dependence as requested by reviewer 1). We extended the spectral range of our measurements, which clearly shows that the background originates from an overlap of the wing of the A exciton signal with that of the B exciton. Knowing this, we are now able to fit the spectra and remove the A exciton background, which drastically reduces the error bar.

6. Comment

*The interesting inverted point in Fig. 3b should be expanded on with additional points in the curve (specifically does the inverted polarization disappear as the pump energy is increased for 2.05 eV?) This would highlight if the predicted *trend* is observed, rather than just a single point.*

Response

We agree with the reviewer that it would be helpful to perform a systematic measurement depending on pump energy in a wider range for the valley polarization of the B exciton. Unfortunately, we are not able to perform degenerate pump-probe measurements in our collinear experimental setup, which prevents us from covering the same range as for the A exciton. However, as suggested by the reviewer, we have measured an additional pump energy of 2.05 eV. In this measurement, the pump laser is already very close to the long-pass filter used to separate the pump laser from the detected probe laser. This causes leakage of pump light into the pump-probe signal. While this leakage can be removed in the data processing, it complicates the fitting of the spectra, which is crucial for the accurate extraction of the valley polarization degree. This makes it complicated to compare the valley polarization degrees directly. However, we can clearly see the inverted valley polarization in the temporal series (Figure S5(c) and (d)).

7. Comment

The discussion of Fig. 3e in the text was confusing. The discussion in p. 5 below Eq. 2 explicitly states that calculations are being discussed, but at parts implies that experimental data is being analyzed, using words like excite and observe. If there are indeed experimental measurements (beyond the upconversion referenced in Ref. 18), these must be better highlighted and presented. If there are not experiments for the CW PL, then the discussion following Eq. 2 should make this clear. The

comparison to Ref. 18 data is good, but why is there no observed polarization of A_{2s} reported in this experiment? Additional CW PL data to support the entire picture of Fig. 3e would be more convincing. Is this CW experiment available to test the entire prediction of CW PL?

Response

We thank the reviewer for bringing up this point. We have improved the revised manuscript and have clarified which part refers to experiments and which to theoretical calculations. We have not performed any CW experiments. All experimental data on CW excitation has been taken from reference [18]. Here, we show that our theoretical calculations on excitation and absorption processes can capture the experimental findings of Ref. [18]. The valley polarization of the A_{2s} exciton in reference [18] is 22% for a pump fluence of 5 μW 29% for 10 μW and 28 for 50 μW. The pump fluence dependence of the valley polarization of the 2s exciton does not follow a clear trend in the experiments of Ref. [18].

In our theory we include the following density-dependent effects: (i) Pauli-Blocking (absorption bleaching via phase space filling), (ii) energetic renormalizations of the Rabi frequency and bandgap (energetic shifts of eigenstates), (iii) intra-valley up-conversion induced by Coulomb scattering channels (non-linear inter excitonic oscillation transfer), and (iv) Dexter-like intervalley couplings induced by Coulomb scattering (linear and non-linear intervalley oscillation transfer). In the case of the pump fluence dependence of the luminescence signal of the 2s resonance our theory captures the intravalley up-conversion observed in the experiment and the intervalley transfer of the Dexter-like interaction. In agreement with the experiment of Ref. [18] our theory predicts a strong up-conversion from the pumped 1s to the 2s state. Since also the Dexter coupling becomes more efficient for higher pump fluences both effects counteract each other for the valley polarization. Intravalley up-conversion favors the 2s signal in the pumped valley, while Dexter coupling favors the 2s signal of the unpumped valley. However, the experiment shows a valley polarization between 20-30%, which is well below our theoretical calculations. This difference may be ascribed to two points: (i) In our theory we simulate the optical response of a perfect crystal with perfect circular dichorism showing the intrinsic limit of valley polarization. (ii) For stronger pump fluences higher-order correlations and Auger scattering channels become important, which may influence the valley polarization. In conclusion, our theory captures the intrinsic limit for low to moderate pump fluences and even describes well the qualitative trends for the strong excitation regime.

In the revised manuscript we have added a discussion on the valley polarization of the A_{2s} exciton.

8. Comment

The Methods provide an outline of the calculations performed, but it is not overly detailed on the procedure. The authors may consider including more numerical details in the Supplementary on how wavefunctions and coupling parameters are calculated and what numbers are used as inputs, which would aid someone in reproducing these calculations.

Response

We have added a new paragraph in the supplementary material, which clarifies the theoretical model in more detail.

Reviewer 3

1. Comment

In this manuscript, the authors have presented a joint theory-experiment study to show the decay of spin and valley polarization in transition metal dichalcogenides (TMDs). This present work reveal a new Coulomb driven intervalley coupling mechanism, that determines the optically accessible spin valley polarization in TMDs. The interaction resembles Dexter coupling between K and K valleys. In general, these results add some new information about monolayer TMDs and could be of interest to specialists in the field, however I doubt that the presented results are novel enough to be published in Nature Communications. I recommend redirecting this manuscript to another more specialized journal.

Response

We thank the referee for carefully reading our manuscript and for his/her general positive evaluation of the new Coulomb driven intervalley coupling mechanism. However, we do not agree that the results are only interesting for specialists in the field. The presented novel Dexter-like mechanism provides the intrinsic limit on the optically accessible valley polarization in the rapidly developing research field of transition metal dichalcogenides. Two-dimensional semiconductors are studied by interdisciplinary research groups to achieve novel spin and valleytronics devices. The intrinsic limit for spin and valley polarization is therefore of profound interest for the large and interdisciplinary community of 2D materials and related heterostructures. Moreover, the novelty of the coupling mechanism gives new insights into fundamental excitation processes, which are of interest for even

broader research communities.

2. Comment

The authors should specifically mention if figure 2 represents experimental data or theoretical modelling. I assume this is theoretical modelling. Further its unclear if the data demonstrates a pump-probe based measurement or a time resolved PL polarization intensity (at zero delay). Moreover the polarization should be maximum at time = 0 fs. Why the peak is shifted from zero?

Response

We thank the reviewer for drawing our attention to this important point. We have clarified the difference between experiment and theoretical calculations in the revised manuscript. The former Fig. 2 has been moved to the supplementary material and is now Fig. S6. while Fig. 4 is now Fig. 5 in the revised manuscript. Both are pure theoretical simulations. This seems to have been a misunderstanding. In Fig. S6 we show the calculated microscopic polarization amplitude not the pump pulse itself. The polarization amplitude rises as long as the pump is present. Therefore, the maximum amplitude is reached only when the pump pulse has vanished and hence after $t=0$. Only in the case of strongly off-resonant excitation or strongly damped excitations the polarization would follow immediately the pump pulse and show its maximum at zero delay time.

3. Comment

2) The authors assume the homogeneous width of the "B" exciton = 50 meV (PAGE 3). I believe they found this number assuming homogeneous width of "A" exciton =10 meV [from Ref. 15]. Since the absorption data in the supplemental material exhibits the linewidth of the "B" exciton feature is 5 times broader than the "A" exciton linewidth, thus homogeneous width of "B" exciton = 50 meV. This approach is incorrect. The linewidth obtained from the absorption is usually governed by inhomogeneous distribution of exciton frequencies. Even at low temperature, the intrinsic homogeneous width is masked by the inhomogeneity. One need to employ either Four wave mixing [e.g. PRL 116, 127402 (2016)] or two dimensional fourier transform spectroscopy [e.g. Nature Communication, 6, 8315, 2015] to estimate the homogeneous width correctly in TMDs.

Response

We agree that a part of the broadening is governed by inhomogeneous broadening. This is particularly important at low temperatures, where the inhomogeneous broadening can dominate the spectral width. To clarify this, we have performed new microscopic calculations to determine the homogeneous broadening of the A and B excitons induced by radiative and non-radiative channels (Nature Commun. 7, 13279 (2016)). The latter are dominated by exciton-phonon scattering in the investigated low-excitation regime. We find that the B exciton exhibits a clearly larger linewidth due to the presence of more relaxation channels. Here, in particular the scattering into the Gamma-hole and the K-Lambda' excitons is important. These are energetically below the B exciton and can be efficiently scattered into via K and M phonons, respectively. We have found a linewidth of 12 meV for the A exciton and 32 meV for the B exciton. We have performed new calculations on the Dexter intervalley coupling explicitly taking into account the influence of the calculated broadening.

4. Comment

Figure 3b exhibits only one experimental data point against the theoretical modelling. Authors need to show more experimental data.

Response

We have performed new experiments and calculations to clarify the importance of the discussed Dexter-like coupling mechanism. We have studied both the temporal evolution of the inverted valley polarization and the effect of the pump intensity on the Dexter-induced carrier transfer. We find that both studies strongly support our model and have significantly improved the revised manuscript, see also comments 2+3 reviewer 1 and reviewer 2.

5. Comment

The authors should explain their experimental details more elaborately. For example: In DTS measurements, what is the pump pulse width? Where the (250 meV) probe pulse is peaked at? Details regarding laser sources (for both pump and probe), and their repetition rates are also need to be mentioned.

Response

As suggested by the Referee, we have added a paragraph in the supporting information explaining the experiment in more detail: The pump-probe experiments are performed with a laser system based on a femtosecond-pulsed fiber laser with a wavelength of 1550 nm and a repetition rate of 40 MHz. Two pulse trains (pump and probe) of the laser are converted into near infrared (NIR)

supercontinua using highly nonlinear fibers. The NIR supercontinua are converted into visible radiation by second harmonic generation (SHG) (Moutzouris et al., Optics Letters 31, 1148 (2006)). The pump pulses have a spectral bandwidth of 8 meV and are peaked at the denoted energy. For probing the B exciton, the probe pulse has a spectral bandwidth of approximately 250 meV and is centered at 2.32 eV, while for probing the A exciton it is peaked at 1.98 eV.

6. Comment

I fail to understand the purpose of introducing the case of MoS₂ (PAGE 3) where the authors predict 25% indirect absorption. This diverts the focus as the authors have neither exhibited any experimental or theoretical data from MoS₂.

Response

The shown simple model can help estimating the intrinsically achievable valley polarization for different TMD materials and understanding the Dexter-like mechanism in general. We have now included more information in the supplementary material containing the estimation of the Dexter coupling elements of excitons for different TMDs. We have also emphasized the dependence on material parameters, such as spin-orbit coupling, lattice constants, the excitonic Bohr radius and the substrate's dielectric constant.

7. Comment

The dielectric constant (ϵ) for SiO₂ is generally varies between 3.9 and 4.0. Is there any specific reason that the authors used $\epsilon = 2.1$ for their calculation (in order to estimate the coupling strength in PAGE 2)?

Response

In contrast to conventional materials, where the static limit of the dielectric constant is used, TMDs exhibit a strong Coulomb interaction, which leads to exceptionally high exciton binding energies and rapid exciton formation. In this case, it is more appropriate to use the high-frequency dielectric constant (2.1 for SiO₂) [(Bechstedt, F. Many-Body Approach to Electronic Excitations; Springer, 2015) + (Nano Lett. 2016, 16, 5568) + (Nano Lett. 2016, 16, 7054)]. Note however, that in the revised manuscript we have performed new calculations considering that in the experiment boron silicate has been used as a substrate, with a dielectric constant of $\epsilon = 4.6$.

8. Comment

In equation 1, what are the coefficients (Γ) signifies? Please explain.

Response

We thank the referee for pointing out this missing definition. These are geometry coefficients, which account for the lattice symmetry. They weigh different Coulomb contributions depending on band and valley of the involved carriers. We have included a new paragraph on the details of the theoretical model to the supplementary material.

Reviewers' comments:

Reviewer #1 (Remarks to the Author):

The authors have included new experimental data and significantly improved the manuscript. I think that the ultrafast (sub ps scale) nature of their observed valley inversion is now quite clear and answers most of the questions I had concerning other possible mechanisms.

In this current version, I consider this manuscript to be suitable for publication.

Reviewer #2 (Remarks to the Author):

The authors have provided responses to the technical comments raised in the previous review. The revised manuscript with added discussion attempts to address the issues raised. Viewed as a whole, I believe that the experiment includes valuable theoretical and experimental insights for the field. There is some question as to the robustness of their model in explaining the data, but the concerns are limited by experimental difficulties which cannot be overcome at the present time. That limitation does not mean the data here are not valuable, though it does suggest room for future work. In recent years, the temporal evolution of valley polarization in transition metal monolayers has been revealed to be a complex issue, and this paper adds insights. As in my original review, I think this work can be published, and the authors have improved their manuscript. The topic is interesting to many researchers in these popular materials who focus on temporally resolved dynamics, although if the relevant community is sufficiently broad for Nature Communications is an editorial judgement with clear room for disagreement.

Some additional suggestions related to the previous technical comments:

5. The authors argue that they are able to reduce the error bar by improving their analysis methods. Since the uncertainty analysis and background issues are somewhat subtle, I think these procedures should be described in the manuscript, Methods, or SI.

6. The additional data point supports the inverted B polarization, but it still misses a key feature of the model expressed in Fig. 3b: the reversal of B exciton VP. In fact, figure 3a also predicts a sign reversal for the A exciton, but the data clearly does not show this. The authors argue that experimental limitations of filtering prevent more B exciton data in the region of interest, which is reasonable, though it implies that the features of their model are not currently testable (or said another way, the model the authors present does not fully capture the salient features of the data). The authors appropriately acknowledge this in their discussion of Fig. 3, and it is not a showstopper as long as the discrepancies are clearly described.

Reviewer #3 (Remarks to the Author):

The authors have addressed all my concerns and the improved manuscript showcases the combination of both theoretical and experimental data which would advance the existing literature on spin-valley physics of transition metal dichalcogenides. I recommend its publication without delay.

We thank all three reviewers for their clear recommendation to publish our work in Nature Communications. Below you find the response to the remaining comments from the second reviewer.

Reviewer #2 (Remarks to the Author):

1. Comment:

The authors argue that they are able to reduce the error bar by improving their analysis methods. Since the uncertainty analysis and background issues are somewhat subtle, I think these procedures should be described in the manuscript, Methods, or SI.

Response:

We agree with the reviewer and have added a paragraph to the supplementary material addressing in more detail the calculation of the uncertainty bars.

2. Comment:

The additional data point supports the inverted B polarization, but it still misses a key feature of the model expressed in Fig. 3b: the reversal of B exciton VP. In fact, figure 3a also predicts a sign reversal for the A exciton, but the data clearly does not show this. The authors argue that experimental limitations of filtering prevent more B exciton data in the region of interest, which is reasonable, though it implies that the features of their model are not currently testable (or said another way, the model the authors present does not fully capture the salient features of the data). The authors appropriately acknowledge this in their discussion of Fig. 3, and it is not a showstopper as long as the discrepancies are clearly described.

Response:

The referee is right - the A-exciton VP is inverted in the theoretical calculation and although the experimental measurement shows a clear reduction, it does not reach the inversion. We have further emphasized the discrepancies between experiment and theory in the manuscript, when explaining the results of Fig. 3 (page 4).